# On the Anisotropic Milling Machinability of DD407 Single Crystal Nickel-Based Superalloy

**DOI:** 10.3390/ma15082723

**Published:** 2022-04-07

**Authors:** Jun Qiu, Tao Yang, Ziyuan Zhang, Qiang Li, Zixin Yan, Libiao Wang

**Affiliations:** 1School of Intelligent Manufacturing, Taizhou University, Taizhou 318000, China; qiujun@tzc.edu.can (J.Q.); zy686@tzc.edu.cn (Z.Z.); 2Zhejiang Provincial Key Laboratory for Cutting Tools, Taizhou University, Taizhou 318000, China; yangtaochd@163.com; 3School of Mechanical Engineering, Liaoning Technical University, Fuxin 123000, China; neuliqiang@163.com; 4School of Pharmaceutical and Materials Engineering, Taizhou University, Taizhou 318000, China; y18969932964@163.com

**Keywords:** DD407 single crystal Ni-based superalloy, milling machinability, anisotropic characteristics, surface roughness, chip edge burr

## Abstract

The DD407 single crystal Ni-based superalloy with a face-centered cubic structure exhibits strong anisotropic characteristics. In order to reveal the material chip formation mechanism and the impact effect of crystal orientations on the materials’ milling machinability, a combination of experimental observations and theoretical analysis were applied in this study. Considering the resolved shear stress and slip system theories, a fundamental theoretical explanation of the milling force and surface quality along different crystal directions on the (001) crystal plane of the DD407 single crystal Ni-based superalloy was proposed based on a previously constructed anisotropic milling model. Our work in this research verifies that [110] crystal direction on the (001) crystal plane of the DD407 single crystal Ni-based superalloy is the most optimal feeding direction during milling, taking into account surface roughness and morphology, slot bottom plastic deformation, work hardening, and chip edge burr feature.

## 1. Introduction

With the rapid development of the aerospace industry and nuclear engineering, engineering materials need to satisfy the higher performance requirements under harsh conditions. With its excellent combined resistance to fatigue, creep, and corrosion under high-temperature and pressure conditions, nickel-based superalloys have attracted a lot of attention. [1,2]. For single crystal Ni-based superalloys, there is no damage caused by slippage along the crystal boundary compared to polycrystalline Ni-based superalloys. Its excellent synthetic mechanical performances of yield strength, structure stability, service reliability, and oxidation and corrosion resistance render them the most potential material in the aerospace industry and nuclear engineering [3]. However, the high strength and low thermal conductivity of the material on a macroscopic scale as well as the presence of hard point particles and high hardening rates on a microscopic scale lead to machining difficulties [4,5], especially in the milling process. The breakage of the material generates violent cutting forces that accelerate tool wear and failure. Although the material preparation method has been developed already, there still is a challenge in the precision machining and semi-finishing of single crystal Ni-based superalloys to satisfy the requirements of high precision and productivity [6,7]. According to the traditional material mechanics theory, the removal of polycrystalline alloy materials is triggered by shearing and slippage along the crystal boundary [8]. However, the machining of single-crystal materials do not conform to this rule due to the absence of grain boundaries and their anisotropic characteristics in mechanics and thermoplastics [9]. In the milling of single crystal Ni-based superalloys, both crystal orientation and crystal plane pose an influence on the crystalline mechanical properties and consequently impact the milling machinability and surface integrity.

In the previous literature, preliminary theory and research were developed on the anisotropic characteristics of single-crystal material machining already. Firstly, the numerical simulations method had contributed many significant results [10,11,12,13]. Chavoshi et al. [14] investigated the anisotropic cutting behavior and found that (111) crystal plane is the most optimal for the cutting of single-crystal silicon. Zhang et al. [15] explored the influences of the cutting speed and depth on the subsurface defect evolution. The experimental results present that the cutting depth can significantly affect the dislocation formation in the subsurface and the crystal structures of the defect layer. Lin et al. [16] constructed tensile deformation stress–strain curves of single-crystal aluminum at nanoscale and proposed the plastic deformation and ductile shear fracture mechanism induced by the multiple slips on the (111) crystal plane. Liu et al. [17] established a single-crystal copper micro-cutting finite element (FE) model and proposed cutting machinability and surface integrity research methods relevant to the anisotropic crystallographic structure. Except for the numerical simulations, experimental studies have also been conducted to investigate the machinability of single-crystal materials in the past years. Based on the crystal plasticity theory and shear strain-based criterion, Ren et al. [18] carried out a series of single-crystal silicon micro-grinding experiments and constructed the relationships between the processing parameters and tool wear conditions on the machining quality. Considering the anisotropic mechanical properties of single-crystal sapphire, Cheng et al. [19] performed massive micro-grinding experiments along different crystal orientations and developed a new grinding force model, which showed that the (0001) orientation could produce a smaller grinding force than the (11–20) orientation. Kota et al. [20] established the mapping relationships between surface quality of single-crystal aluminum after ultraprecision turning and cutting force as well as crystal orientations. Cai et al. [21] deduced the shear and elastic modulus for the (001) crystal plane of a Ni-based single crystal superalloy along different crystal orientations. Considering the actual geometric shape of the abrasive embedded in the grinding rod, Zhou et al. [22] constructed the grinding force model and verified the model accuracy through a series of experiments. Then, they discussed the effects of the micro-grinding parameters and material anisotropic characteristics on the grinding force of Ni-based single crystal superalloys. These studies greatly improve our understanding both scientifically and technologically. Nonetheless, it is imperative to further study DD407 anisotropic characteristics in detail during the milling process for its industrial applications.

In this work, a series of milling experiments were designed and conducted along different crystal directions on (001) crystal plane of the DD407 Ni-based superalloy. The milling machinability was systemically investigated, including the milling force, surface roughness, and morphology as well as work hardening. In addition, chip edge burr characteristics during milling were also studied by combined experimental observation, theoretical analysis, and FE simulations. Considering the above aspects, the best tool feed direction was recommended for the milling of the DD407 Ni-based superalloy.

## 2. Methodology

### 2.1. Milling Experiments

The DD407 superalloy used in this work was prepared with the seed crystal method and grew along the [001] crystal orientation (The composition can be referred to Table 1). To research the effects and mechanism of the tool feeding directions on DD407 milling machinability, different crystal orientations should be indicated prior to milling as shown in Figure 1. Firstly, a fan block was cut from the DD407 sample in which the axial direction is [001] orientation, as shown in Figure 1a. In order to eliminate the change of cutting direction caused by the oblique machining interpolation algorithm of the CNC machine tool, the machining direction is consistent with the crystal direction. Samples in the direction of [100], [010], [470], and [110] were cut into four rectangular blocks, and the size of the sample block is 15 × 6 × 6 mm^3^. In the process of cutting force testing, the straight line milling along the Y-axis is centered to ensure a scientific and rigorous cutting force testing process. For the fan block, [100] direction was detected as a reference in Figure 1b, according to the directional cutting method for face-centered cubic (FCC) crystalline materials proposed in the previous literature [23]. The included angle is 17.97° between the [100] direction and the edge of the rectangle block. Finally, the other three directions of 45°, 60°, and 90° with the [100] crystal direction were assigned to [110], [470], and [010] crystal directions, as depicted in Figure 1c.

All milling experiments were carried out on a four-axis milling complex machining center (TC-S2Cz/-O, Brother, Nagoya, Japan) displayed in Figure 2, which has a maximum spindle rotational speed of 6000 rpm. Integral end milling tools of GM-4E coated with TiAlN by PVD were chosen in this study. The milling speed (vs), tool feed per tooth (fz), and milling depth (ap) employed in the experiments were 37.7 m/min, 0.013 mm, and 0.18 mm, respectively. The milling experiments were carried out along different directions of [010], [470], [110], and [100] on the (001) plane. During milling of the DD407 superalloy, minimal quantities of lubricant (MQL) cooling method were adopted with a YSHT-MQL2551A (Foshan, China) supplier of dual atomized nozzles, and the cooling oil is KS1106 MQL (Shanghai, China)oil supplied by KINS. The nozzle tilt angle to horizontal and spray distance were set as 30° and 80 mm, respectively. The air pressure and oil droplet flow rate were kept at 0.5 MPa and 80 mL/h, respectively.

Machined surface morphology, chip edge burr characteristics, subsurface plastic deformation, and work hardening properties were explored by a Hitachi S-4800 (Tokyo, Japan)scanning electron microscope, Zeiss Axio Scope A1 (Oberkochen, Germany)metallographic microscope, and Shimazu HMV-2T (Kyoto, Japan)microhardness tester, respectively. The loading condition of the hardness test is 0.100 Kg. In addition, Olympus OLS50003D (Tokyo, Japan)confocal microscope was adopted to detect the machined microsurface profile and roughness.

### 2.2. Finite Element Model of Slot Milling

To investigate stress and temperature distributions in DD407 under milling, a FE cutting model was constructed. During the milling process, the cutting motion can be decomposed into the linear motion along the feeding direction and the rotary motion around the spindle. The force and motion analysis processes were simplified based on the end mill helix angle feature, as shown in Figure 3.

For the purpose of applying the milling model to the oblique shear model in the Abaqus/Explicit environment, Coulomb’s law and the Johnson–Cook constitutive model were adopted. An 8-node linear brick, reduced integration, hourglass control methods were selected to define the element. The metal cutting process is a multi-factor coupling mechanical model and the factors such as strain rate and thermal softening need to be considered simultaneously [24]. The Johnson–Cook model is used extensively in metal cutting because of its comprehensiveness
(1)$$\sigma =(A+B\varepsilon^{n})\left[1+C\ln\left(\frac{\dot{\varepsilon }}{{\dot{\varepsilon }}_{0}}\right)\right]\left[1-{\left(\frac{T-T_{r}}{T_{m}-T_{r}}\right)}^{m}\right]$$
where *A*, *B*, *n*, *C*, and *m*, are material constants and $T_{r}$, $T_{m}$, σ, ε, $\dot{\varepsilon }$, and ${\dot{\varepsilon }}_{0}$ are the room temperature, transition temperature, equivalent flow stress, equivalent plastic strain, equivalent plastic strain rate, and reference strain rate, respectively. The input parameters are given in Table 2.

The Johnson–Cook model is used for material failure. This model is based on the equivalent plastic strain value at the element integral point. The failure process can be divided into two stages: damage failure and damage evolution. The stress–strain relationship between these two stages can be described as:(2)$$\bar{\varepsilon_{f}^{pl}}=\left[d_{1}+d_{2}\exp(d_{3}\frac{p}{q})\right]\left[1+d_{4}\ln(\frac{\bar{\varepsilon^{pl}}}{\varepsilon_{0}})\right]\left[1+d_{5}\hat{\theta }\right]$$
where $d_{1}\text{–}d_{5}$ represent the failure constants and p, *q*, $\varepsilon^{pl}$ and $\hat{\theta }$ are the compressive stress and mises stress, plastic strain rate, reference temperature respectively. Refer to Table 3 for $d_{1}\text{–}d_{5}$.

In the finite element model, damage parameter *D* is a measurement factor to judge element failure (element delete). This parameter is based on a cumulative law that is defined as:(3)$$D=\sum \frac{\Delta \varepsilon^{p}}{\varepsilon^{f}}$$
where $\Delta \varepsilon^{p}$ is the equivalent plastic strain increment and $\varepsilon^{f}$ is the failure strain. When *D* = 1, the element is deleted.

## 3. Results and Discussion

### Cutting Force

Figure 4 depicts milling forces along different crystal directions on the (100) plane of the DD407 superalloy. It can be seen that the milling forces fluctuated in a range of 15.23 to 47.97 N in the X direction and 9.64 to 20.83 N in the Z direction, feeding along crystal directions designed in this experiment. The milling force along the [110] crystal direction is the lowest among all directions and gradually increased along the [110] to [100] crystal directions as well as [110] to [010] crystal directions.

The difference in the milling forces when feeding along three families of crystal direction is mainly derived from the critical stress for the activation of material deformation, which is called Peierls–Nabarro stress (*P*−*N* stress). The *P*−*N* stress, *τP*−*N*, can be expressed as:(4)$$\tau_{P-N}=\frac{2G}{1-v}\exp\left[-\frac{2\pi d_{hkl}}{\left(1-v\right)b}\right]$$
where, *G*, *dhkl*, *v,* and *b* are the shear modulus, the spacing of (*hkl*) plane, the Poisson’s ratio, and the magnitude of Burgers vector, respectively. Due to the crystal direction of [110], [010], [100], and [470], the values of *d_hkl_* and *b* caused by material deformation are different, leading to the variation of the milling forces. However, the difference in the milling forces among the same family of crystal direction is determined by the stress decomposition regulation. For example, between [100] and [010], though they have the same *P*−*N* stress, the stress decomposition follows
(5)$$\tau =\tau_{P-N}\cdot cos\theta \cdot cos\varphi$$
where, *τ* is the real stress caused by material deformation, *θ* is the included angle between the milling force and the glide plane, *φ* is the included angle between the milling force and the glide direction. Though the values of the *P*−*N* stress and *θ* are the same for two crystal directions in the same family, the difference of *φ* will also make the milling force different from each other.

Figure 5 and Figure 6 show morphological profiles and the measured surface roughness of the slot bottoms machined along different crystal directions. From the curve of the roughness vs. crystal direction, it can be seen that it has a similar trend to the curve of the milling force in Figure 4. The surface roughness of [110] is the best, while the surface roughness of [100] is the worst. The surface roughness after milling depends on the shear modulus in the feeding direction. For the milling on the (100) plane of FCC materials, the shear modulus can be written as [23]:(6)$$G_{100}=\frac{1}{S_{44}+\left[\left(S_{11}-S_{12}\right)-\frac{1}{2}S_{44}\right]sin^{2}2\theta }$$where Sq (root mean square height), Ssk (skewness), Sku (kurtosis), Sp (maximum peak height), Sv (maximum pit height), Sz (maximum height), Sa (Arithmetical mean height), Sdq (root mean square slope), and Sdr (developed area ratio) can be described by Equations (7)–(13), respectively, and which Ref. to ISO-25178. The *A* is the length of measurement.
(7)Sq=1A∬AZ2x, ydxdy(8)Ssk=1Sq31A∬AZ3x, ydxdy(9)Sku=1Sq41A∬AZ4x, ydxdy(10)Sp=maxzx,y(11)Sv=minzx,y(12)Sz=Sp+Sv(13)Sa=1A∬AZx, ydxdySdq=1A∬A∂zx,y∂x2+∂zx,y∂y2dxdySdr=1A∬A1+∂zx,y∂x2+∂zx,y∂y2−1dxdy
where, S11, S12, and S44 are the flexibility coefficients of the DD407 superalloy, θ is the included angle between feeding direction and [100] crystal direction in (100) plane. From Figure 1, it can be calculated that the values of θ for the crystal directions of [100], [110], and [470] are 0°, 45°, and 60°, respectively. According to Equation (6), the shear modulus along [100] crystal direction is the largest compared to that of [110] and [470] crystal directions, making material removal difficult during milling. Therefore, surface quality is the worst along [100] crystal direction and became better successively when turning the feeding direction to [470] and [110]. When feeding along [110] direction, the lowest surface roughness was achieved.

As shown in Figure 7a, the hardness of the four directions in descending order is [100], [010], [470], and [010]. In addition, the fluctuation of [100] direction is significant, while the difference in data fluctuation between [470] and [010] is not obvious. Apart from the surface roughness, microstructure in the subsurface also plays a crucial role in the performance of the DD407 superalloy after milling.

In order to further explore the mechanism of the hardness difference, the cross-sectional metallographic images milled along different crystal directions were observed after corrosion, as shown in Figure 7b. In virtue of its single-crystal characteristics, the DD407 superalloy will not exhibit typical plastic deformation phenomena in the metallographic images, such as crystal boundary torsion and grain elongation. However, surface properties of the DD407 superalloy, including hardness and corrosion resistance of the subsurface, maybe change under the combination of thermomechanical coupling and alternating load during milling. In the metallographic image of DD407 milling along [110] crystal direction, there is no obvious metamorphic layer. Therefore, [110] has the minimum hardness in macroscopic physical properties.

When feeding along [100] crystal direction, a discontinuous metamorphic layer of about 4.3 μm in depth appeared in the subsurface as depicted in Figure 7b, arising from the thermal and mechanical coupling action during milling. With further turning the milling direction into [470] and [010] crystal direction, the metamorphic layer became continuous and not uniform in depth. The evolution of microstructure in the subsurface will absolutely pose an influence on its mechanical performance.

To reveal the underlying mechanism of hardness variation in the subsurface after milling [27], finite element modeling (FEM) by a commercial Abaqus software was adopted to simulate the strain field distribution under maximum feed per tooth. From the FEM analysis results presented in Figure 8, it can be observed that the simulated thickness of the strain-affected layer is 3.6 μm. Therefore, it is reasonable to conclude that the strain hardening effect mainly impacts the subsurface with several micrometers, while the thermal softening effect dominates the hardness in the subsurface with larger depth, resulting in the effect of “hardening-softening-hardening” displayed in Figure 9.

As shown in Figure 10, during the milling of single crystal DD407 superalloy, cracks generated on the chip side due to adiabatic shear action, forming chip edge burr. The formation of chip edge burr will aggravate the tool wear and thus shorten the tool life. Therefore, it is critical to investigate the fundamental formation mechanism of cracks for the purpose of optimizing the cutting conditions.

Considering the oblique cutting model, the chip motion vector is not along the normal direction to the main cutting edge on the cutting plane, induced by the existence of helix angles and junk shots. This will make the interaction and relative motion appear between chips and the rack face of the tool, resulting in massive residual tensile stress accumulated in the chip matrix along the cutting depth direction. It can be clearly observed that plastic deformation and flow occurred during the adiabatic shear process of the chip.

The deformation will further introduce lamellar structures in the chips and grooves on the chip-free surface. When the residual tensile stress exceeded the material fracture strength, cracks generated on the groove side and expanded along the longitudinal direction of the chip, forming longitudinal chip edge burrs. Furthermore, as the milling speed increased, the plastic flow was intensified for the materials on both sides of the crack, leading to the morphology of chip edge burr transforming from rectangle/trapezoid to triangular structure.

However, on the chip constrained side, the plastic flow was inhibited by the machined surface, so that there is no obvious chip edge burr formation except for a small crack, as illustrated in Figure 11. To elucidate the formation of the transverse crack, a simplified model of oblique cutting was established in Figure 11. In the theoretical model, taking account of the influence of the end mill helix angle, the cross-section of the chip was set as a right-angled trapezoid rather than a rectangle. The cutting force, Fresultant which was induced by the interaction between the rake face and the chip, keeps constant and is along the direction normal to the chip-free side (plane ABCD).

Component force perpendicular to plane CDC’D’, F1, can induce the stress concentration on the groove side and trigger the generation of longitudinal crack. At the beginning stage of the chip edge formation, F1 was much larger than F2. However, the concentrated stress was released with the longitudinal crack formation, causing F1 to decrease greatly but F2 to increase sharply. When the stress perpendicular to plane A’B’C’D’ exceeded the material fracture strength, cracks formed along the chip formation direction.

In order to reveal the detailed process of the chip edge burr generation for single crystal DD407 superalloy during milling, FE calculation was also performed to analyze the stress field evolution. For Stage I, the cutting edge began to participate in milling and the removed chip flowed steadily along the tool rake face.

The stress in the cutting zones was uniformly distributed along the cutting depth direction without concentrated stress. Therefore, the chip morphology is kept complete without crack generation. At Stage II, the cutting edge had entirely entered the cutting zone, and the removed material moved along the oblique upper direction of the tool rake face. In the cutting zone, stress concentration can be detected. The residual stress on the chip surface gradually decreased with the increasing distance to the machined surface, producing massive micro-cracks near the chip-free side. At Stage III, with the expansion of the micro-cracks, the concentrated stress in the chip-free side was released and reached the chip constrained side. Moreover, obvious plastic flow appeared on both sides of the cracks, leading to the geometric structure of the crack gradually transforming into a triangle. Finally, at Stage VI, the first crack with a triangle shape formed completely, and the second chip edge began to generate near the main cutting edge.

## 4. Conclusions

Previously, the milling machinability of metallic materials was commonly considered to be isotropic. Therefore, the cutting direction can not be considered a key research object. The main contribution of this paper is to reveal that the milling machinability of DD407 is anisotropic. In this study, DD407 single crystal Ni-based superalloy samples were milled on the (001) crystal plane feeding along four different directions of 0°, 45°, 60°, and 90° with the [100] crystal direction, corresponding to the [100], [110], [470], and [010] crystal directions. Anisotropic milling machinability of the DD407 single crystal Ni-based superalloy was estimated by investigating the milling force, surface roughness, and morphology, work hardening, chip edge burr feature by the combination of experimental observation, theoretical analysis, and FE simulations. The results demonstrate that, when feeding along the crystal direction of [110] on the (001) crystal plane, the milled DD407 single crystal Ni-based superalloy possesses the lowest surface roughness and the best surface quality in all feeding directions. The smallest cutting force obtained in the milling along the crystal direction of [110] can also be beneficial to prevent tool wear. In addition, the work hardening in the subsurface and the formation of the chip edge burr were analyzed using FE simulations.

## Figures and Tables

**Figure 1 materials-15-02723-f001:**
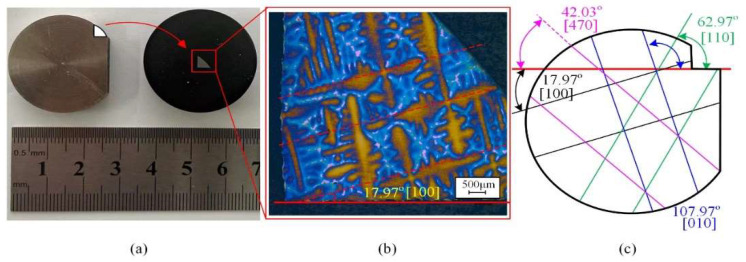
DD407 directional cutting method. (**a**) Directional block, (**b**) DD407 microstructure, (**c**) Specific crystal directions designation on (001) plane of DD407.

**Figure 2 materials-15-02723-f002:**
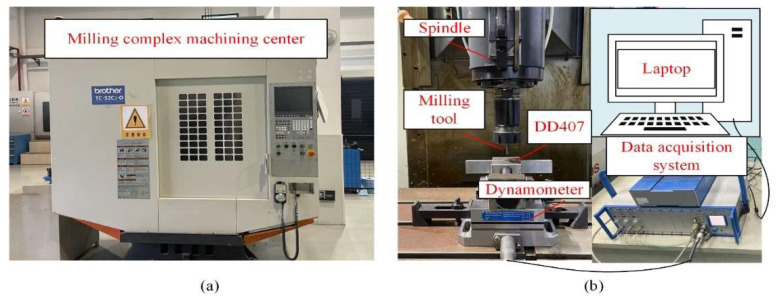
(**a**) The milling complex machining system and (**b**) the text system.

**Figure 3 materials-15-02723-f003:**
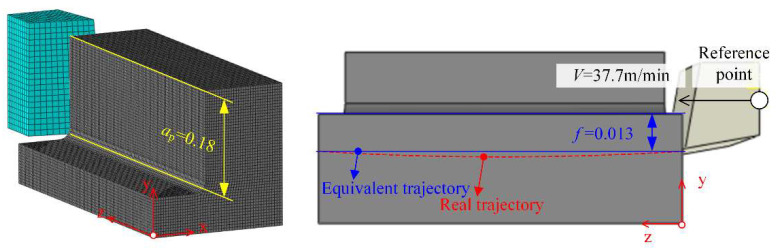
Oblique shear model of the milling process.

**Figure 4 materials-15-02723-f004:**
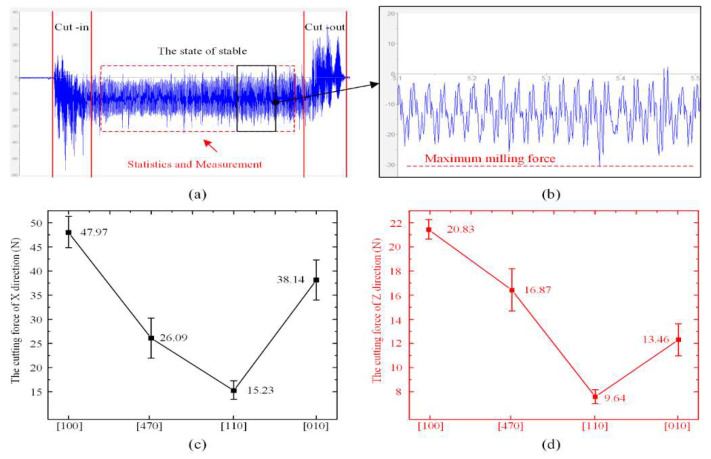
(**a**) Unprocessed cutting force data, (**b**) Maximum milling force, (**c**) the cutting force of X, and (**d**) Z direction.

**Figure 5 materials-15-02723-f005:**
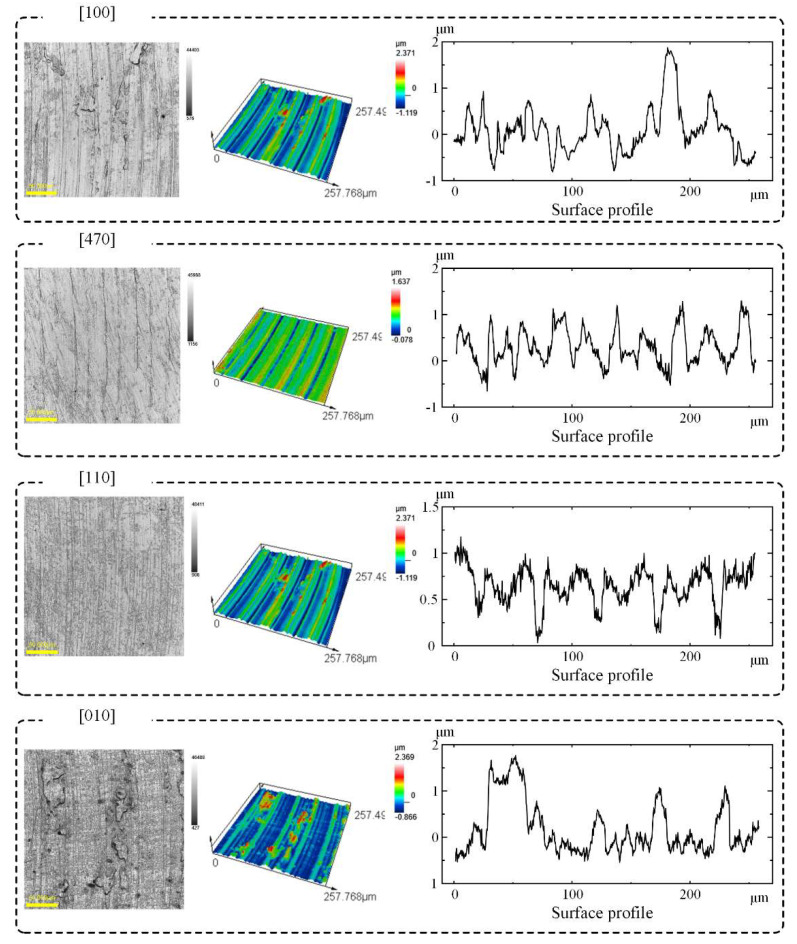
Surface roughness Sa of slot bottoms milled along different crystal directions.

**Figure 6 materials-15-02723-f006:**
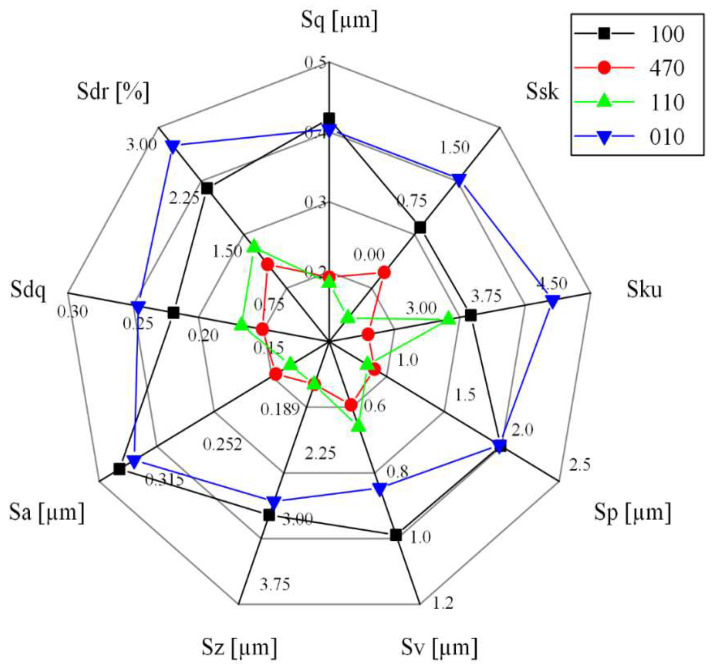
Surface Roughness of slot bottoms milled along different crystal directions.

**Figure 7 materials-15-02723-f007:**
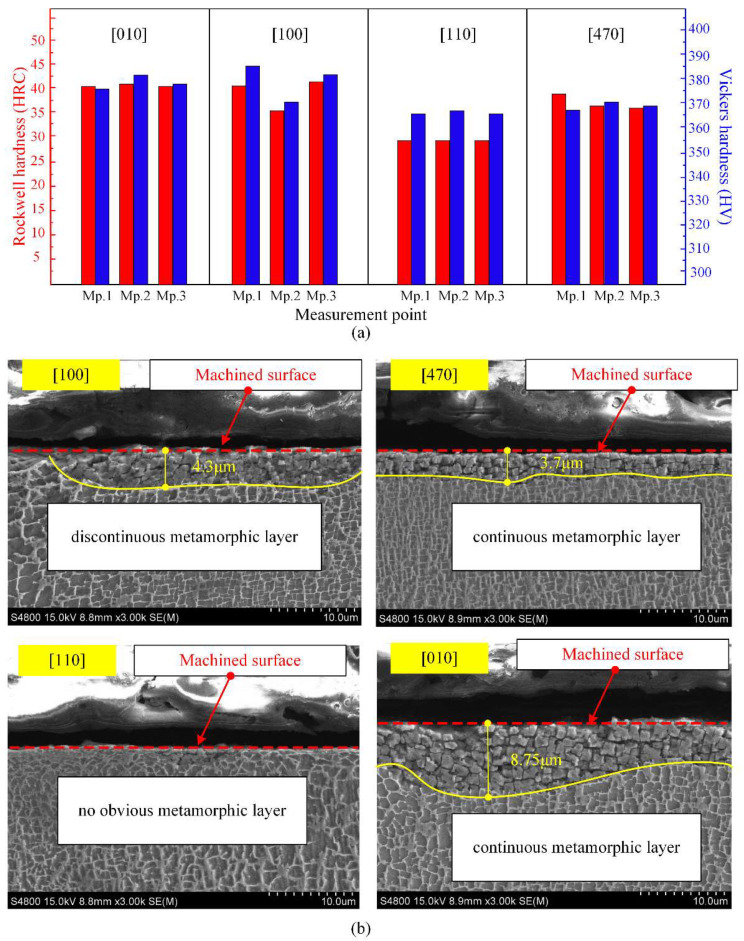
(**a**) Hardness test results and (**b**) subsurface metamorphic layer.

**Figure 8 materials-15-02723-f008:**
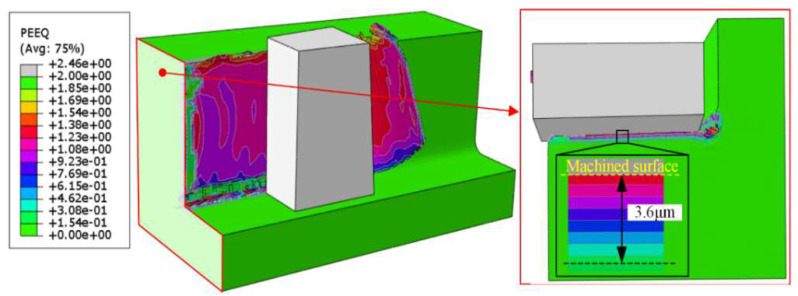
Distributions of strain in the milling of DD407.

**Figure 9 materials-15-02723-f009:**
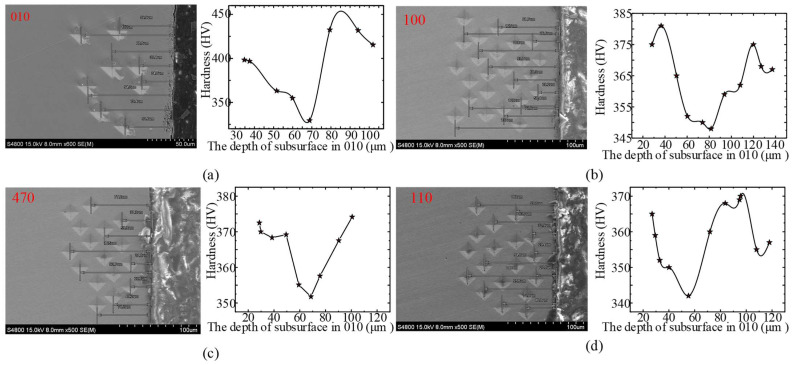
The depth of measurement points and the relationship between hardness and the depth of subsurface in (**a**) 010, (**b**) 100, (**c**) 470, and (**d**) 110.

**Figure 10 materials-15-02723-f010:**
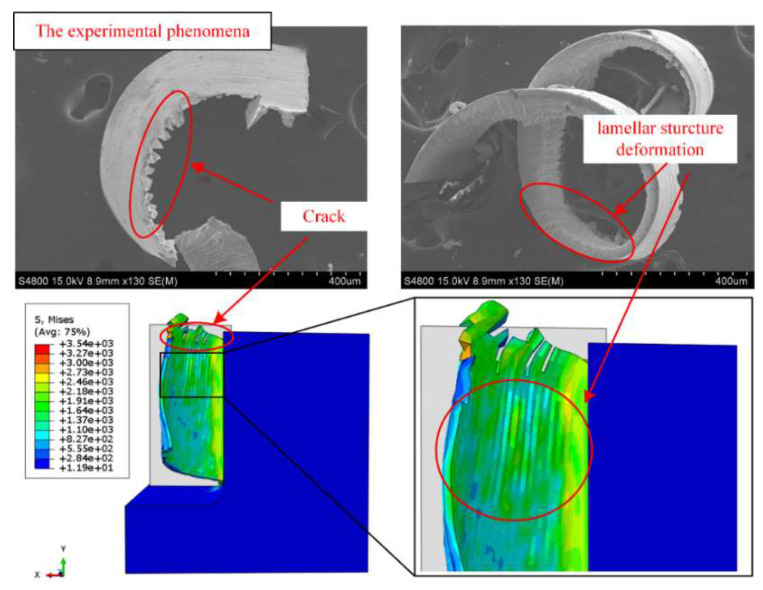
Cracks and lamellar structure were found in experiments and simulations.

**Figure 11 materials-15-02723-f011:**
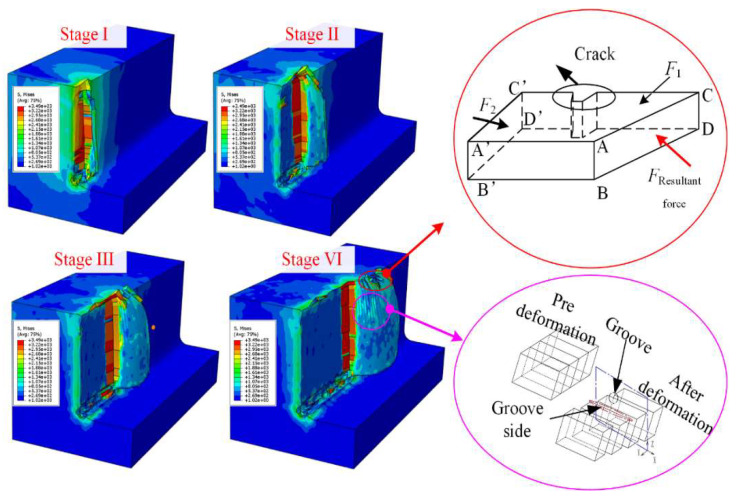
The chip edge crack and lamellar structure formation mechanism.

**Table 1 materials-15-02723-t001:** The element distribution of DD407.

Element	C K	Al K	Si K	Ti K	Cr K	Co K	Ni K	W M
Weight percentage (%)	9.33	5.79	1.1	1.6	7.68	5.67	64.19	4.63
Element percentage (%)	32.01	8.85	1.62	1.38	6.09	3.97	45.06	1.04

**Table 2 materials-15-02723-t002:** Johnson–Cook material parameters [25].

A (MPa)	B (MPa)	C	n	m	$T_{m}\;({}^{\circ}C)$	$T_{r}\;({}^{\circ}C)$	$\dot{\varepsilon_{0}}\;(s^{-1})$
1562	300	0.0164	0.25	1.7	1200	25	1

**Table 3 materials-15-02723-t003:** Material damage parameters of Johnson–Cook [26].

$d_{1}$	$d_{2}$	$d_{3}$	$d_{4}$	$d_{5}$
0.11	0.75	−1.45	0.04	0.89

## Data Availability

Not applicable.

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
