# Peer review of "On the Anisotropic Milling Machinability of DD407 Single Crystal Nickel-Based Superalloy"

_materials, 2022, doi:10.3390/ma15082723_

Round 1

Reviewer 1 Report

he work is devoted to the study and modeling of the machinability of a single crystal in different crystallographic directions. The work is of great scientific and technological interest and can be published after minor shortcomings are eliminated. There are a number of comments on the work:

  1. What do the coefficients A, B, C, m иn in equation 1 mean and where did they come from
  2. Did the authors use the same shear modulus in different crystallographic directions (equation 4)?
  3. «The surface roughness after milling depends on the shear mod- 198
    ulus in the feeding direction.» Authors should comment on this thesis
  4. How do the obtained results in different planes agree with the atomic packing density in these planes?

Author Response

We express our sincere appreciation to the reviewer for the careful reading of our work and constructive comments. It is our hope that the revised manuscript has addressed all the comments raised by the reviewers and provides a clearer and more informative description of our work. The answers and explanations for all comments are listed in the attachment.

Reviewer 2 Report

The work is devoted to an interesting topic and the results obtained can find their reader.

At the same time, there are a number of comments that I consider it necessary to share.

1. It is doubtful that this article fits the Topical Collection "Alloy and Process Development of Light Metals". Nickel alloys are usually not classified as light metals.

2. In the text, it would be advisable to decipher the composition of DD407 single crystal Ni-based superalloy at least once. At least this way - DD407(Ni-7.82Cr-5.34Co-2.25Mo-4.88W-6.02Al-1.94Ti-3.49Ta in wt.%) or something similar.

3. The text of the article should be more accurately formatted. In particular, in formulas 1 and 2 - an unnecessary sign &. In the list of authors, affiliation links are not presented as indexes and the like.

4. I would like the paper to end with broader conclusions / generalizations. Why should these results be of interest to those who work with alloys other than DD407? You should to show it.

Author Response

(The authors gave the same response as above.)

Reviewer 3 Report

The manuscript describes the study of the anisotropic milling machinability of DD407 single crystal nickel-based superalloy. The research work is well proposed, realized and clearly presented. The reviewer suggests accepting the paper with minor revisions.

The reviewer’s comments/remarks which should be incorporated in the manuscript:

  1. The parameters of Johnson-Cook model are taken from the reference no. 23. The material damage parameters are taken from different reference no. 24. The authors should provide explanation of such a choice of the parameters. In other word a defined set of parameters of JC model doesn’t correspond to the material damage parameters. It’s a bit suspicious as the damage parameters of JC model are dependent on the plasticity model of JC.
  2. The parameters of material damage of Johnson-Cook in Table 2 are taken from the reference No. 24. However, those parameters aren’t directly quoted in this reference. Please, provide better reference and clear description/identification of the material damage parameters.
  3. Please, add the orientation axes (X,Y,Z) according to the FEM in the Figure 3.
  4. Please, describe or introduce in the manuscript the parameters of the surface roughness from the Figure 6 (Sq, Ssk, Sku,…).
  5. Please, correct the description of the direction in the Figure 7a- hardness results.
  6. Why is the hardness of milled surfaces measured by Rockwell method (Figure 7) and the hardness of the subsurface area by Vickers method (Figure 8)? The authors should provide clear explanation. In the case of Vickers method the loading level is missing (it’s seems to be microhardness level).
  7. In the case of the hardness measurement of the milled surfaces (Rockwell method) subsurface region in analyses and thus can be rather influenced by the orientation of the single crystal. It would worth to provide the hardness in these single crystal orientations for comparison.
  8. Line 2. Please, correct the Title of the work. There’s missing space (“MillingMachinability”).
  9. Line 92. Please, correct the upper index (“mm3”).
  10. Line 138. Please, correct the symbols in the equation 1.
  11. Line 142. Please, delete the dot (“Table. 1”).
  12. Line 150. Please, correct the symbols in the equation 2.
  13. Line 152. Please, delete the dot (“Table. 2”).
  14. Line 176. Please, correct the English in the sentence “For the crystal…”

Author Response

(The authors gave the same response as above.)

Reviewer 4 Report

In the manuscript titled "On the Anisotropic MillingMachinability of DD407 Single 2 Crystal Nickel-based Superalloy" the authors studied the effect of machinability parameters on the DD407 single crystal superalloy along with different directions. The manuscript can be accepted with minor revision.

  1. There are some grammatical errors that need to be corrected.
  2. The introduction does not describe the previous work done on the machinability of single crystal Ni-based superalloys. Authors need to add detailed literature to provide a background.
  3. How does present work add advancement to the already available information?

Author Response

(The authors gave the same response as above.)
